EMBO
Molecular Medicine

# RIG-I antiviral signaling drives interleukin-23 production and psoriasis-like skin disease

Huiyuan Zhu[1], Fangzhou Lou[1], Qianqian Yin[1], Yuanyuan Gao[1], Yang Sun[1], Jing Bai[1], Zhenyao Xu[1], Zhaoyuan Liu[1], Wei Cai[1], Fang Ke[1], Lingyun Zhang[1], Hong Zhou[1], Hong Wang[1], Gang Wang[2], Xiang Chen[3], Hongxin Zhang[4], Zhugang Wang[4], Florent Ginhoux[5], Chuanjian Lu[6], Bing Su[1,3] & Honglin Wang[1,6,7,*] iD

## Abstract

Retinoic acid inducible-gene I (RIG-I) functions as one of the major sensors of RNA viruses. *DDX58*, which encodes the RIG-I protein, has been newly identified as a susceptibility gene in psoriasis. Here, we show that the activation of RIG-I by 5′ppp-dsRNA, its synthetic ligand, directly causes the production of IL-23 and triggers psoriasis-like skin disease in mice. Repeated injections of IL-23 to the ears failed to induce IL-23 production and a full psoriasis-like skin phenotype, in either germ-free or RIG-I-deficient mice. RIG-I is also critical for a full development of skin inflammation in imiquimod (IMQ)-induced psoriasis-like mouse model. Furthermore, RIG-I-mediated endogenous IL-23 production was mainly confined to the CD11c[+] dendritic cells (DCs) via nuclear factor-kappa B (NF-κB) signaling, and stimulated RIG-I expression in an auto-regulatory feedback loop. Thus, our data suggest that the dysregulation in the antiviral immune responses of hosts through the innate pattern recognition receptors may trigger the skin inflammatory conditions in the pathophysiology of psoriasis.

**Keywords** DCs; interleukin-23; NF-κB; psoriasis; RIG-I
**Subject Categories** Immunology; Skin

## Introduction

Recent studies have highlighted the pathogenic role of the IL-23/Th17 axis in psoriasis, a chronic inflammatory skin disorder affecting the skin in 1–3% of the general population (Nestle *et al*, 2009; Lowes *et al*, 2013). IL-23 has been demonstrated as a key master cytokine that promotes T helper 17 (Th17) cell survival and proliferation, and targeting the IL-23p19 subunit with a monoclonal antibody results in the clinical improvement in psoriasis (Zheng *et al*, 2007; Kopp *et al*, 2015). However, in psoriatic lesions, IL-23 is also a potent antiviral factor by promoting the survival and proliferation of Th17 cells that release IL-29 to simulate the secretion of antiviral proteins by psoriatic keratinocytes (Lowes *et al*, 2013; Wolk *et al*, 2013). In the skin, IL-23 is mainly produced by tissue-resident and/or recruited dendritic cells (DCs), and to a lesser extent, by keratinocytes (Lee *et al*, 2004; Piskin *et al*, 2006). Yet what triggers IL-23 production in the skin of individuals genetically predisposed to develop psoriasis is still unclear. *DDX58* encoding the RIG-I protein has been suggested as a psoriasis susceptibility gene (Nair *et al*, 2009). RIG-I was initially reported as an innate immune receptor, which acts as a cytoplasmic sensor of viral RNA-binding protein and plays a major role in the activation of a cascade of antiviral responses including the induction of type I interferons and the activation of the nuclear factor-kappa B (NF-κB) signaling to produce proinflammatory cytokines (Loo & Gale, 2011).

Psoriasis can be exacerbated or provoked by a variety of different environmental factors, including infections. So far, what infection(s) triggers psoriasis in genetically predisposed population remains poorly understood. Here, using IL-23- and imiquimod (IMQ)-induced psoriasis-like mouse model, we sought to investigate whether and how the RIG-I-mediated antiviral signaling is involved in the production of IL-23 and the development of psoriasis-like skin lesions.

Here we show that RIG-I expression is markedly increased in the affected skin derived from psoriasis patients and from both IL-23- and IMQ-induced psoriasis-like mouse model. The activation of RIG-I by 5′ppp-dsRNA directly causes the activation of NF-κB and the subsequent production of IL-23 and triggers cutaneous inflammation

---

1  Key Laboratory of Cell Differentiation and Apoptosis of Chinese Ministry of Education, Department of Immunology and Microbiology, Shanghai Institute of Immunology, Shanghai Jiao Tong University School of Medicine (SJTU-SM), Shanghai, China
2  Department of Dermatology, Xijing Hospital, The Fourth Military Medical University, Xi'an, China
3  Department of Dermatology, Xiangya Hospital, Central South University, Changsha, China
4  Research Centre for Experimental Medicine of Rujin Hospital, Shanghai Jiao Tong University School of Medicine, Shanghai, China
5  Singapore Immunology Network (SIgN), Agency for Science, Technology and Research (A*STAR), Singapore City, Singapore
6  Guangdong Provincial Hospital of Chinese Medicine, Guangdong Provincial Academy of Chinese Medical Sciences, The Second Clinical School of Guangzhou University of Chinese Medicine, Guangzhou, China
7  Shanghai Key Laboratory for Tumor Microenvironment and Inflammation, Shanghai Jiao Tong University School of Medicine (SJTU-SM), Shanghai, China
   *Corresponding author. Tel: +86 21 63846590 ext. 776727; Fax: +86 21 64660996; E-mail: honglin.wang@sjtu.edu.cn

in mice that closely resembles human psoriasis. We demonstrate that repeated injections of IL-23 to the ears are not able to induce IL-23 production and a full psoriasis-like skin phenotype, in germ-free and RIG-I knockout mice, and RIG-I is also critical for a full development of skin inflammation in IMQ-induced psoriasis-like mouse model. Furthermore, we identify that RIG-I-mediated endogenous IL-23 production is mainly confined to the CD11c$^+$ dendritic cells (DCs) via NF-κB signaling, and stimulates RIG-I expression in an auto-regulatory feedback loop. Therefore, our data indicate that the RIG-I-mediated antiviral response is pivotal for the IL-23 production in the DCs and for the IL-23- and IMQ-induced psoriasis-like skin inflammation in mice, highlighting that the dysregulation in the antiviral immune responses of hosts through the innate pattern recognition receptors may initiate the chronic skin inflammation in psoriasis.

# Results

## Germ-free mice injected with IL-23 fail to develop psoriasis-like skin inflammation in ears

To study a potential role of innate immunity in the etiology of psoriasis, we performed repeated injections of IL-23 to the ears of germ-free (GF) mice. We found that the intradermal injection of IL-23 failed to induce an increase in full ear thickness of the mice in GF conditions compared with specific pathogen-free (SPF) conditions (Fig 1A). Hematoxylin and eosin (H&E) staining confirmed that the GF mouse ears administrated with IL-23 showed a robust decrease in epidermal acanthosis and dermal inflammatory cellular infiltration (Fig 1B–D). Furthermore, in contrast to the SPF mice, numbers of Ki67$^+$ cells were significantly decreased in the GF mice treated with IL-23, indicating that the excessive proliferation of basal keratinocytes induced by IL-23 was reduced in the absence of microbiota (Fig 1E and F). The innate immune system is the first line of defense to combat invading microbes via the pattern recognition receptors including the Toll-like receptors, the NOD-like receptors, the C-type lectin receptors, the RIG-I-like receptors, and cytosolic DNA sensors (Hornung et al, 2009; Takeuchi & Akira, 2010). Because the RIG-I protein is encoded by *DDX58,* which has been newly identified as a susceptibility gene in psoriasis (Nair et al, 2009), we here set up experiments to study the role of RIG-I in psoriatic pathology in the IL-23-induced mouse model of psoriasis.

The RIG-I-mediated antiviral response leads to activation of the NF-κB, which is a major proinflammatory pathway involved in psoriasis (Loo & Gale, 2011). Western blot analysis revealed markedly decreased RIG-I expression that corresponded to the diminished levels of phosphorylated IκBα in the IL-23-treated ears of the GF mice compared with those of the SPF mice, suggesting that IL-23 is capable of triggering the expression of RIG-I and the activation of NF-κB in SPF mice, but not in GF mice (Fig 1G). Moreover, we found that repeated injections of ears with IL-23 induced a profound increase of IL-23 and IL-17 protein levels in the SPF mice, whereas the IL-23 injection slightly triggered IL-23, but not IL-17 expression in the GF mice (Fig 1H and I). IL-23 is a crucial proinflammatory factor in psoriasis pathogenesis (Kopp et al, 2015). To investigate whether endogenous production of IL-23 is essential for development of psoriasis-like skin inflammation in IL-23-induced mouse

model, we performed repeated IL-23 injections in IL-23$^{-/-}$ mice. We found that IL-23$^{-/-}$ mice exhibited less severity of psoriasis-like skin inflammation compared with wild-type (WT) mice, including ear thickness and other disease features (Fig EV1). Together, our data suggest that the microbiota recognized by the innate immune system is required for RIG-I expression, and the subsequent NF-κB activation and endogenous IL-23 production in IL-23-induced mouse model of psoriasis.

## 5′ppp-dsRNA triggers psoriasis-like skin disease

To specifically investigate the contribution of RIG-I-mediated antiviral signaling in psoriasis-like skin inflammation, we injected 5′ppp-dsRNA (complexed with the cationic lipid LyoVec, 5′ppp-dsRNA) or 5′ppp-dsRNA control (complexed with the cationic lipid LyoVec, dsRNA Ctr) into the ears of wild-type (WT) mice on every second day for 2 weeks. Strikingly, we demonstrated that the injection of 5′ppp-dsRNA, but not dsRNA Ctr, markedly increased the formation of demarcated erythematous plaques (Fig 2A). Correspondingly, the thickness and the acanthosis of ear skin injected with 5′ppp-dsRNA were also significantly increased compared with dsRNA Ctr (Fig 2B–D). Histological analysis of the inflammation revealed a significant increase in dermal cell infiltration in 5′ppp-dsRNA-treated mice compared with dsRNA Ctr-treated controls (Fig 2E). Moreover, we detected that the expression levels of Ki67 in the epidermis were significantly increased in the mouse ears treated with 5′ppp-dsRNA when compared with dsRNA Ctr treatment (Fig 2F and G). Together, these data clearly show that the activation of RIG-I antiviral signaling by its specific synthetic ligand 5′ppp-dsRNA is capable of triggering psoriasis-like skin inflammation.

## Increased RIG-I expression in lesional skin of psoriasis mouse models and patients

By quantitative real-time polymerase chain reaction (qPCR) and immunohistochemistry analysis, we detected that RIG-I expression was significantly increased in lesional skin of patients with psoriasis, and the up-regulation of RIG-I was confined to the epidermal keratinocytes and the dermal inflammatory cells (Fig 3A–C). Consistently, RIG-I expression was markedly enhanced in the affected skin of IL-23-induced model at both mRNA and protein levels (Fig 3D and E). Furthermore, increased RIG-I protein levels were also found in the lesional skin of IMQ-induced psoriasis-like mouse model (Fig 3F). More importantly, we observed that RIG-I expression, as indicated by the histological score (H score), was closely correlated with skin acanthosis in the human psoriatic lesions (Fig 3G and Table EV1). These data suggest that the antiviral signaling mediated by RIG-I is highly activated in the skin lesions of patients and two mouse models of psoriasis.

## RIG-I is required for IL-23- and IMQ-induced psoriasis-like skin inflammation

We next investigated the role of RIG-I in the IL-23-induced model by using RIG-I-deficient (RIG-I$^{-/-}$) mice (Wang et al, 2007). We demonstrated that the deletion of RIG-I led to a pronounced decrease in plaque formation (Fig 4A). Notably, RIG-I$^{-/-}$ mice exhibited a significant decrease in the ear thickness compared with

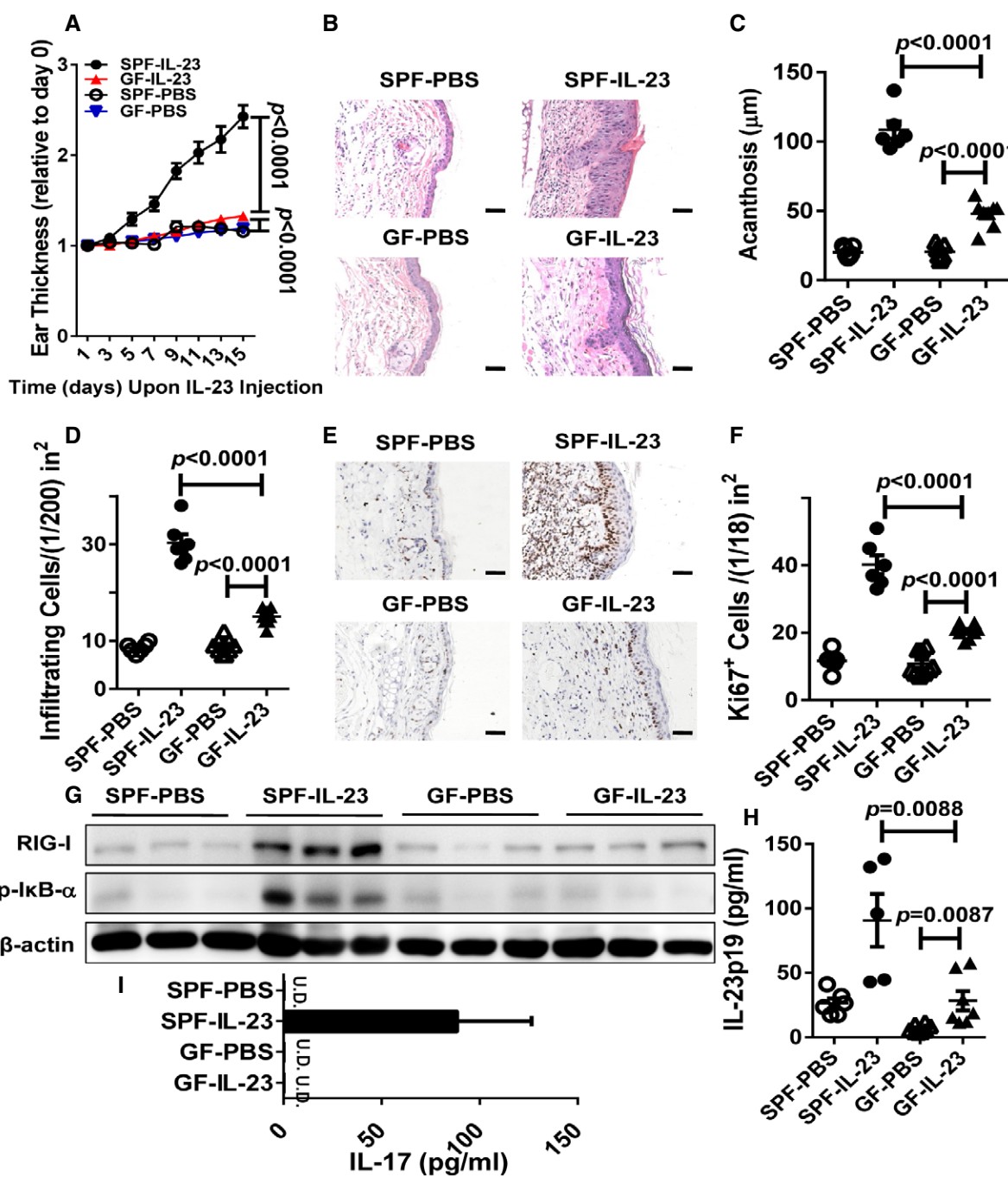

**Figure 1. Germ-free mice are almost protected from IL-23-induced psoriasis-like skin inflammation.**

A   The ear thickness of WT mice in specific pathogen-free (SPF) conditions or in germ-free (GF) conditions treated with PBS or IL-23. Data are presented on the indicated day relative to day 0. Significant differences are indicated: $P < 0.0001$, one-way ANOVA, $n = 6–9$ per group (mean ± SEM).

B   Representative H&E staining of the ears treated as in (A), $n = 6–9$ per group. Scale bar: 50 μm.

C   Acanthosis of the ears treated as in (A). Significant differences are indicated: two-tailed Student's *t*-test, $n = 6–9$ per group (mean ± SEM).

D   Numbers of dermal infiltrating cells of SPF and GF mice treated with PBS or IL-23. Significant differences are indicated: two-tailed Student's *t*-test, $n = 6–9$ per group (mean ± SEM).

E   Representative immunostaining of Ki67 in lesional skin derived from SPF and GF mice treated with PBS or IL-23, $n = 6–9$ per group. Scale bar: 50 μm.

F   Quantitation of Ki67+ cells in ear skin. Significant differences are indicated: two-tailed Student's *t*-test, $n = 6–9$ per group (mean ± SEM).

G   Western blot analysis of RIG-I and phosphorylated IκBα (p-IκBα) expression after treatment as in (A).

H, I   ELISA analysis of IL-23p19 (H) and IL-17 (I) protein levels in supernatants of ear skin homogenates derived from indicated groups. Significant differences are indicated: two-tailed Student's *t*-test, $n = 3–8$ per group (mean ± SEM), U.D., undetected.

Source data are available online for this figure.

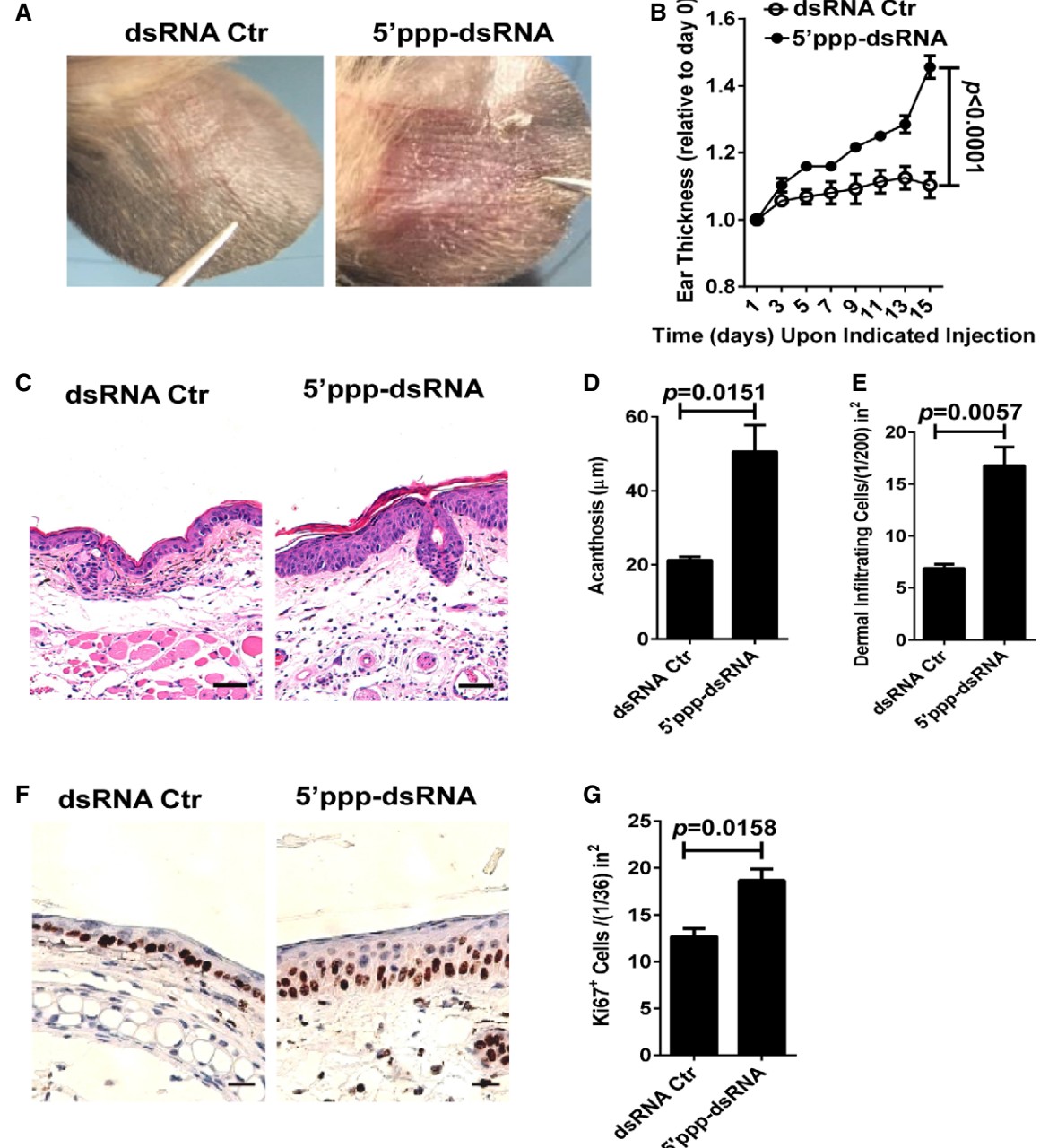

**Figure 2. 5′ppp-dsRNA, a synthetic ligand of RIG-I, triggers psoriasis-like skin inflammation in mice.**

A    Representative photographs of the ears after intradermal injection with indicated administration on every other day for eight times, $n$ = 4–5 per group.

B    The ear thickness of WT mice treated as in (A) on the indicated day expressed relative to day 0. Significant differences are indicated: $P < 0.0001$, one-way ANOVA, $n$ = 4–5 per group (mean ± SEM).

C    Representative H&E staining of the ears treated as in (A), $n$ = 4–5 per group. Scale bar: 50 μm.

D, E    Acanthosis and dermal cellular infiltrates of ears subjected to indicated administration. Significant differences are indicated: two-tailed Student's $t$-test, $n$ = 4–5 per group (mean ± SEM).

F    Representative immunostaining of Ki67 in skin derived from indicated groups, $n$ = 4–5 per group. Scale bar: 20 μm.

G    Quantitation of Ki67$^+$ cells in ear skin. Significant differences are indicated: two-tailed Student's $t$-test, $n$ = 4–5 per group (mean ± SEM).

WT mice (Fig 4B). Moreover, the histological analysis of the inflammation revealed a marked decrease in epidermal hyperplasia (acanthosis) in the RIG-I$^{-/-}$ mice (Fig 4C and D). Numbers of CD3$^+$ T cells and CD11c$^+$ DCs were pronouncedly reduced in the RIG-I$^{-/-}$

mice (Fig EV2A–C). In addition, the expression of Ki67 was also significantly decreased in RIG-I$^{-/-}$ mice compared with the controls (Fig 4E and F). Strikingly, we found that the disruption of RIG-I almost blocked the activation of the NF-κB signaling, and decreased

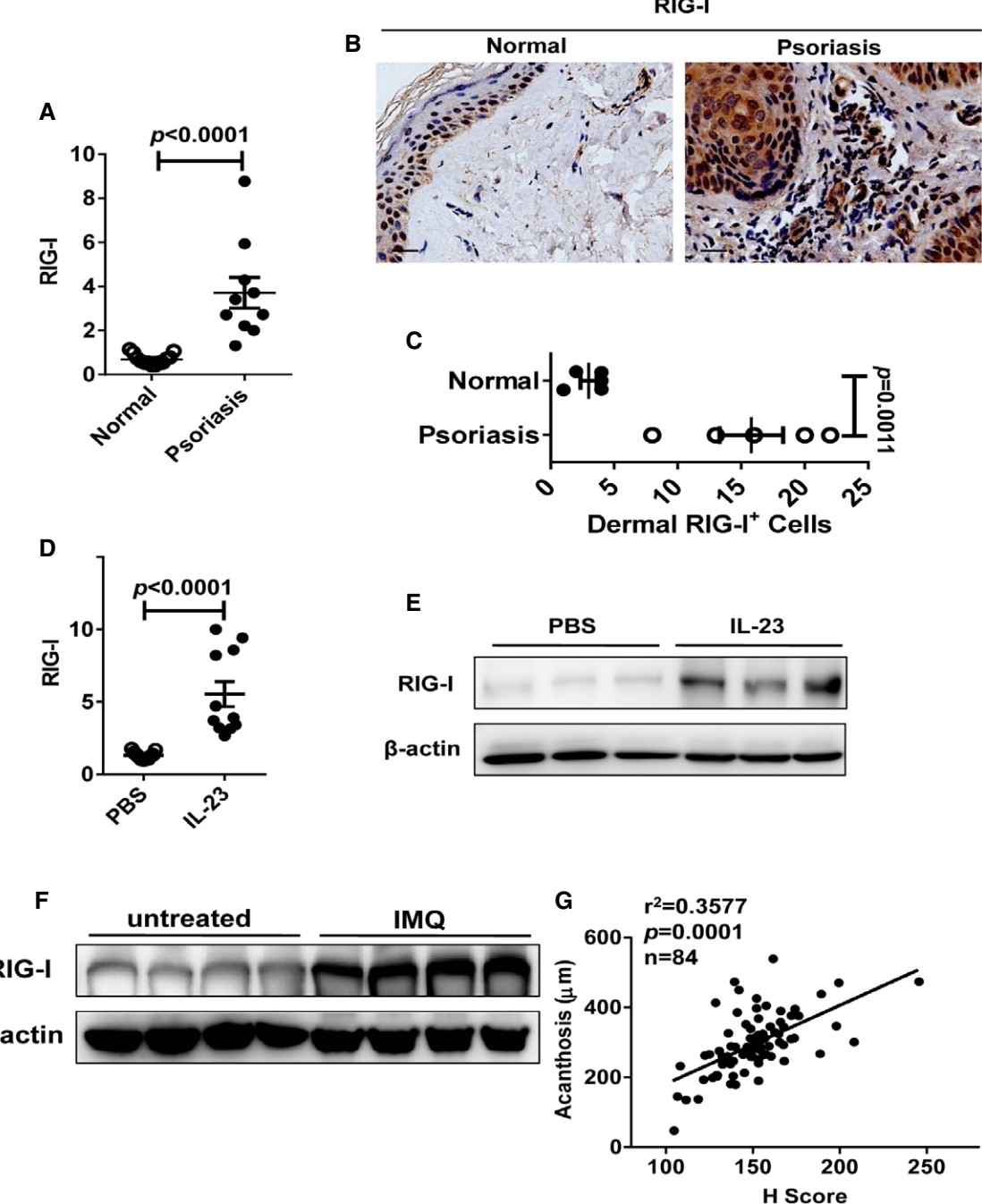

**Figure 3.  RIG-I expression is increased in lesional skin of psoriasis mouse model and patients.**

A   qPCR analysis of RIG-I mRNA expression in psoriatic lesions compared to healthy samples. Significant differences are indicated: two-tailed Student's *t*-test, *n* = 10–13 per group (mean ± SEM).

B   Representative immunohistochemical detection of RIG-I in skin of psoriatic lesions compared with healthy samples, *n* = 5 per group. Scale bar: 20 μm.

C   Quantitation of RIG-I+ cells in dermis of psoriatic skin compared to healthy skin. Significant differences are indicated: two-tailed Student's *t*-test, *n* = 5 per group (mean ± SEM).

D   qPCR analysis of RIG-I mRNA expression in ear skin of mice treated with PBS or IL-23. Significant differences are indicated: two-tailed Student's *t*-test, *n* = 11 per group (mean ± SEM).

E   Western blot analysis of RIG-I expression in ears treated with PBS or IL-23 (*n* = 3).

F   Western blot analysis of RIG-I expression in ears untreated or treated with IMQ (*n* = 4).

G   Correlation between RIG-I expression and acanthosis of 84 psoriatic skin samples (*r*² = 0.3577). Significant differences are indicated: Pearson test, two-tailed.

Data information: qPCR values expressed as the ratio of mRNA to β-actin, relative to control samples, and indicated as fold change.

Source data are available online for this figure.

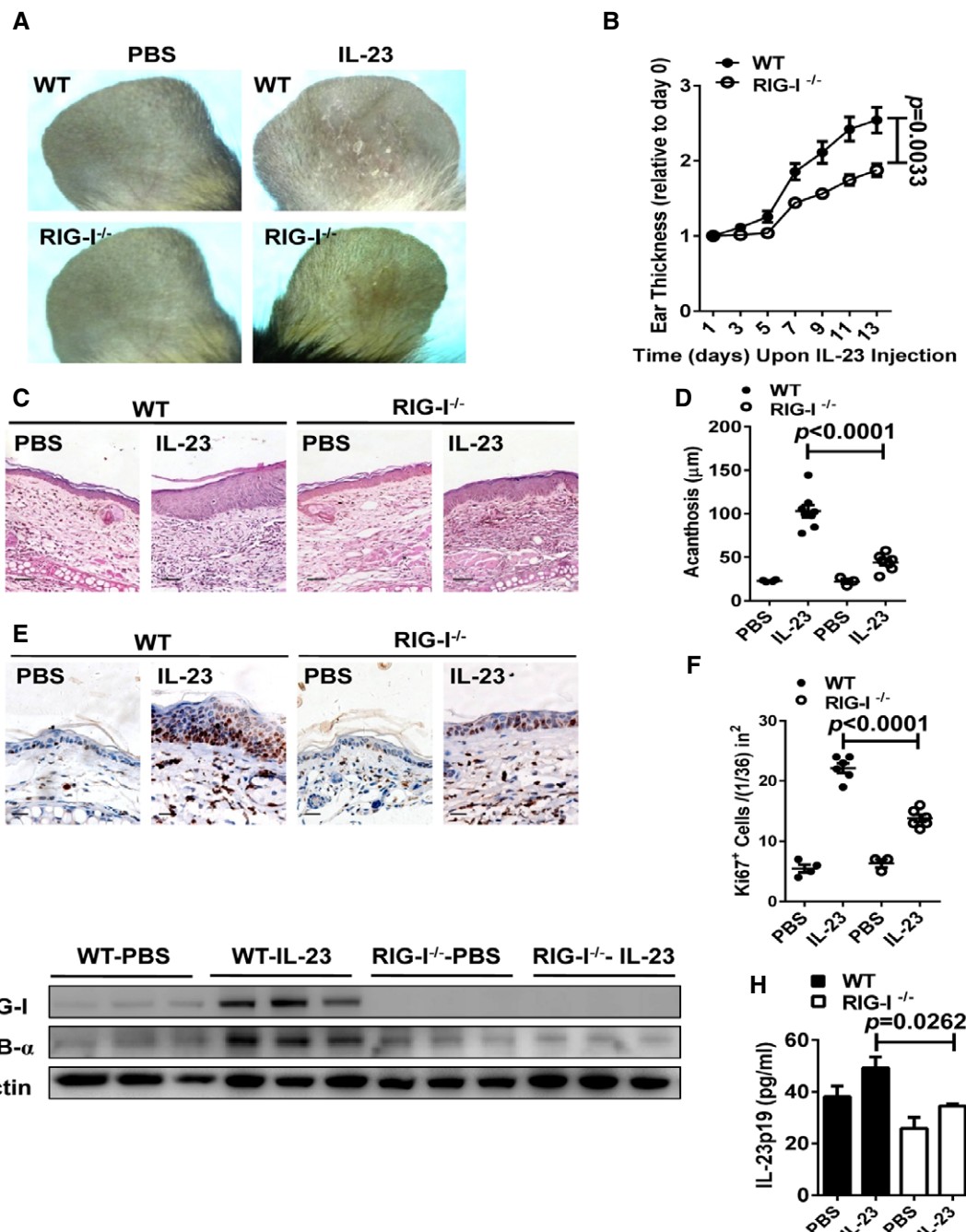

**Figure 4. IL-23-induced psoriasis-like skin disease is attenuated in RIG-I$^{-/-}$ mice.**

A  Representative photographs of the ears of WT mice (upper panel) and RIG-I$^{-/-}$ mice (lower panel) after intradermal injection with PBS or IL-23 (500 ng) on every other day for eight times, $n$ = 4–5 per group.

B  The ear thickness of WT and RIG-I$^{-/-}$ mice on the indicated day presented relative to day 0. Significant differences are indicated: $P$ = 0.0033, one-way ANOVA, $n$ = 4–5 per group (mean ± SEM).

C  Representative H&E staining of the ears treated as in (A), $n$ = 4–5 per group. Scale bar: 50 μm.

D  Acanthosis of WT and RIG-I$^{-/-}$ mice treated with PBS or IL-23. Significant differences are indicated: two-tailed Student's $t$-test, $n$ = 3–8 per group (mean ± SEM).

E  Representative immunostaining of Ki67 in ear skin derived from WT and RIG-I$^{-/-}$ mice treated with PBS or IL-23, $n$ = 3–6 per group. Scale bar: 20 μm.

F  Quantitation of Ki67$^+$ cells in ear skin derived from WT and RIG-I$^{-/-}$ mice treated with PBS or IL-23. Significant differences are indicated: two-tailed Student's $t$-test, $n$ = 3–6 per group (mean ± SEM).

G  Western blot analysis of RIG-I and p-IκBα expression after treatment as in (A).

H  ELISA detection of IL-23p19 protein levels in supernatants of ear skin homogenates derived from indicated groups. Significant differences are indicated: two-tailed Student's $t$-test, $n$ = 3 per group (mean ± SEM).

Source data are available online for this figure.

the production of IL-17 induced by repeated IL-23 injections (Figs 4G and EV2D). Furthermore, we observed a reduced production of IL-23 in the RIG-I$^{-/-}$ ears injected with IL-23 (Fig 4H). Consistently, we observed that the deletion of RIG-I significantly attenuated psoriasis-like skin inflammation in IMQ-induced psoriasis-like mouse model (Fig EV3A–F), and decreased the production of IL-23 and IL-17 in lesional skin induced by IMQ (Fig EV3G and H). Taken together, our data reveal that mice lacking RIG-I display a decreased severity of key disease features including IL-23 and IL-17 production, and highlight that RIG-I might play a pivotal role in the keratinocyte proliferation and the epidermal hyperplasia in IL-23- and IMQ-induced psoriasis-like mouse models.

### RIG-I expression in haematopoietic cells is pivotal for IL-23-induced psoriasis-like skin inflammation

RIG-I is expressed by both the keratinocytes and the dermal inflammatory cells (Fig 3A–C). To further distinguish the critical role of RIG-I in non-haematopoietic skin-resident cells compared with radiosensitive haematopoietic cells in IL-23-mediated skin inflammation, we generated reciprocal bone marrow chimeric mice. Lethally irradiated WT or RIG-I$^{-/-}$ mice were reconstituted with WT bone marrow. Interestingly, the recipients of both genotypes had comparable levels of severity of psoriasis-like skin inflammation (Fig EV4). However, after repeated injections of IL-23, the WT recipient mice displayed a marked decrease in the ear thickness when reconstituted with the RIG-I$^{-/-}$ bone marrow (Fig 5A). Also, acanthosis and Ki67 expression were significantly decreased in the WT recipient mice reconstituted with the RIG-I$^{-/-}$ bone marrow compared with the WT controls (Fig 5B–E). These results suggest that RIG-I likely functions in the haematopoietic cells to regulate the cutaneous pathology.

### Activation of RIG-I in DCs triggers IL-23 production

Stimulation of the innate TLR7/TLR8 toll-like receptors with imiquimod (IMQ), a synthetic agonist is sufficient to trigger IL-23-mediated psoriasis-like skin inflammation (van der Fits et al, 2009), and CD11c$^+$ DCs are the principal source of IL-23 in the IMQ-induced psoriasis-like skin lesions (Riol-Blanco et al, 2014). We next sought to identify the RIG-I-expressing cell type(s) in the dermis in IL-23-induced skin lesions. After the repeated injections of IL-23, the lesions had a significant increase in RIG-I$^+$CD45$^+$ leucocytes compared with the PBS-treated controls (Fig 6A and B). These RIG-I$^+$CD45$^+$ leucocytes were mainly composed of CD11c$^+$ DCs (Fig 6C and D). To confirm the important role of RIG-I in the regulation of the expression of IL-23, we stimulated bone marrow-derived DCs (BMDCs) with IL-23 and/or 5′ppp-dsRNA. We observed that either IL-23 or 5′ppp-dsRNA significantly increased IL-23 mRNA expression in the BMDCs compared with non-treated controls, and the effect of their combination on IL-23 expression was even more remarkable (Fig 6E). However, IL-23 mRNA levels were undetected in the RIG-I$^{-/-}$ BMDCs stimulated with IL-23, 5′ppp-dsRNA, or both (Fig 6E). Moreover, we found that 5′ppp-dsRNA directly triggered the production of IL-23 in WT BMDCs, but not in the RIG-I$^{-/-}$ BMDCs (Fig 6F). In IMQ-induced psoriasis-like mouse model of psoriasis, the selective activation of TLR-7 in the skin CD11c$^+$ DCs caused an increase in IL-23 production and psoriasis-like skin

inflammation (Wohn et al, 2013). By analyzing RIG-I$^{-/-}$ mice, RIG-I was found to play an essential role in antiviral signaling in fibroblast and CD11c$^+$ conventional DCs (Kato et al, 2005). In accordance with these studies, we demonstrate here that the CD11c$^+$ DCs are the principal skin DCs mediating the activation of RIG-I antiviral signaling and subsequent IL-23 production in the IL-23-induced psoriasis-like skin inflammation.

### NF-κB signaling mediates the expression of IL-23 in DCs

The IFN regulatory factor (IRF) 3/7-mediated type I interferon and the NF-κB signaling pathways are the two major pathways downstream of RIG-I activation (Yoneyama & Fujita, 2007). We next characterized the molecular pathways in the RIG-I-mediated production of IL-23 in the DCs. Using specific siRNAs, we silenced p65 and IRF-3/7 in the BMDCs (Fig 7A). In contrast to the controls, the knockdown of p65, but not IRF-3/7, led to a significant decrease in the IL-23 expression in the BMDCs stimulated with IL-23 or 5′ppp-dsRNA (Fig 7B–D), suggesting that the NF-κB signaling pathway is required for the RIG-I-mediated production of IL-23. Database analysis revealed a potential p65 binding site in the promoter of the IL-23p19 subunit (Fig 7E). To verify the binding of the p65 component of NF-κB to the promoter of IL-23p19, we carried out chromatin immunoprecipitation (ChIP) assays. The IL-23p19 promoter abundance was increased by ~2.0-fold in the WT DCs, and 1.4-fold in the RIG-I$^{-/-}$ DCs under the IL-23 stimulation (Fig 7F). A luciferase reporter assay further revealed that p65 binding to the promoter element of IL-23p19 mediated IL-23 expression in BMDCs (Fig 7G). RIG-I initiates signaling pathways culminating in the activation of NF-κB that is constitutively activated in psoriatic lesions and contributes to epidermal hyperproliferation (Gugasyan et al, 2004; Rebholz et al, 2007; Kawai & Akira, 2008). Herein, our data demonstrate that the activation of NF-κB signaling in the downstream of RIG-I is responsible for IL-23 induction in the DCs.

## Discussion

Psoriasis vulgaris is an immune-mediated chronic inflammatory skin disease and can be triggered or exacerbated by various environmental factors, particularly infections. Recent studies revealed that psoriasis was triggered after receiving influenza vaccinations, reflecting the notion that the disease is possibly induced by viruses (Shin et al, 2013; Sbidian et al, 2014; Gunes et al, 2015). In this study, we uncovered a critical innate immune pathway in which RIG-I-mediated antiviral response is critical for triggering the IL-23 expression and the development of psoriatic pathology in psoriasis. Moreover, monoclonal antibodies that bind the IL-23p40 or p19 subunit have demonstrated efficacy in the psoriasis treatment (Papp et al, 2008; Kopp et al, 2015). Herein, we provided evidences that RIG-I is up-regulated in lesional skin of psoriasis patients, and its activation promotes the production of IL-23 in DCs, a cell type known to be important for the initiation of the disease. We further demonstrate that RIG-I expression is induced by IL-23 in an autoregulatory feedback loop. RIG-I exerts function via NF-κB signaling pathway, which is excessively activated in psoriatic lesions and has a close link with psoriasis as discovered by genome-wide association studies (Lizzul et al, 2005; Oka et al, 2012). Thus, our research

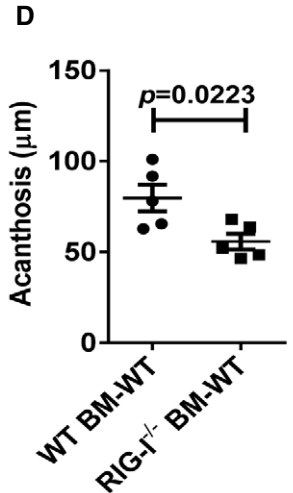

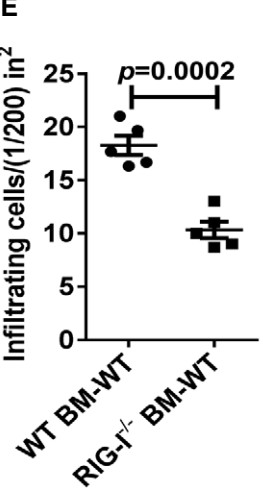

**Figure 5.  RIG-I expression in haematopoietic cells is critical for IL-23-induced psoriasis-like skin inflammation.**

A    Lethally irradiated WT mice were adoptively transferred with WT or RIG-I$^{-/-}$ bone marrow cells, and the generated chimeric mice were subjected to IL-23-induced psoriasis-like skin inflammation. Graph represents the ear thickness of WT recipient mice on the indicated day relative to day 0. Significant differences are indicated: $P < 0.0001$, one-way ANOVA, $n = 5$ per group (mean $\pm$ SEM).

B, C    Representative H&E staining and Ki67 immunostaining of the ears derived from indicated groups, $n = 5$ per group. Scale bar: 50 μm.

D    Acanthosis of WT BM to WT recipients (WT BM-WT) or RIG-I$^{-/-}$ BM to WT recipients (RIG-I$^{-/-}$ BM-WT) treated with IL-23. Significant differences are indicated: two-tailed Student's *t*-test, $n = 5$ per group (mean $\pm$ SEM).

E    Dermal cellular infiltrates of WT BM-WT or RIG-I$^{-/-}$ BM-WT mice treated with IL-23. Significant differences are indicated: two-tailed Student's *t*-test, $n = 5$ per group (mean $\pm$ SEM).

revises the current understanding of psoriasis etiology by showing that RIG-I is a key mediator in psoriasis based on its role in sensing virus infection and promoting IL-23 production.

The initiation of psoriasis is commonly followed by infections, drugs, and environmental stress (Perera *et al*, 2012). Although accumulating evidences have shown that streptococcal throat infection is strongly associated with guttate psoriasis (Valdimarsson *et al*, 2009), which supports the notion that the initiation of psoriasis may be partially due to bacterial infection. Yet, there is no compelling

evidence that anti-streptococcal interventions and tonsillectomy in patients with guttate or chronic plaque psoriasis (psoriasis vulgaris) are beneficial (Owen *et al*, 2000). Interestingly, patients affected by hepatitis C or papilloma virus are associated with induction or exacerbation of psoriasis (Imafuku & Nakayama, 2013; Jain *et al*, 2015). Moreover, stimulation of the antiviral pattern recognition receptors such as TLR7/8 with IMQ, a synthetic agonist, is sufficient to trigger psoriasis-like skin inflammation in mice and psoriasis in humans (Wu *et al*, 2004; Fanti *et al*, 2006; van der Fits *et al*, 2009),

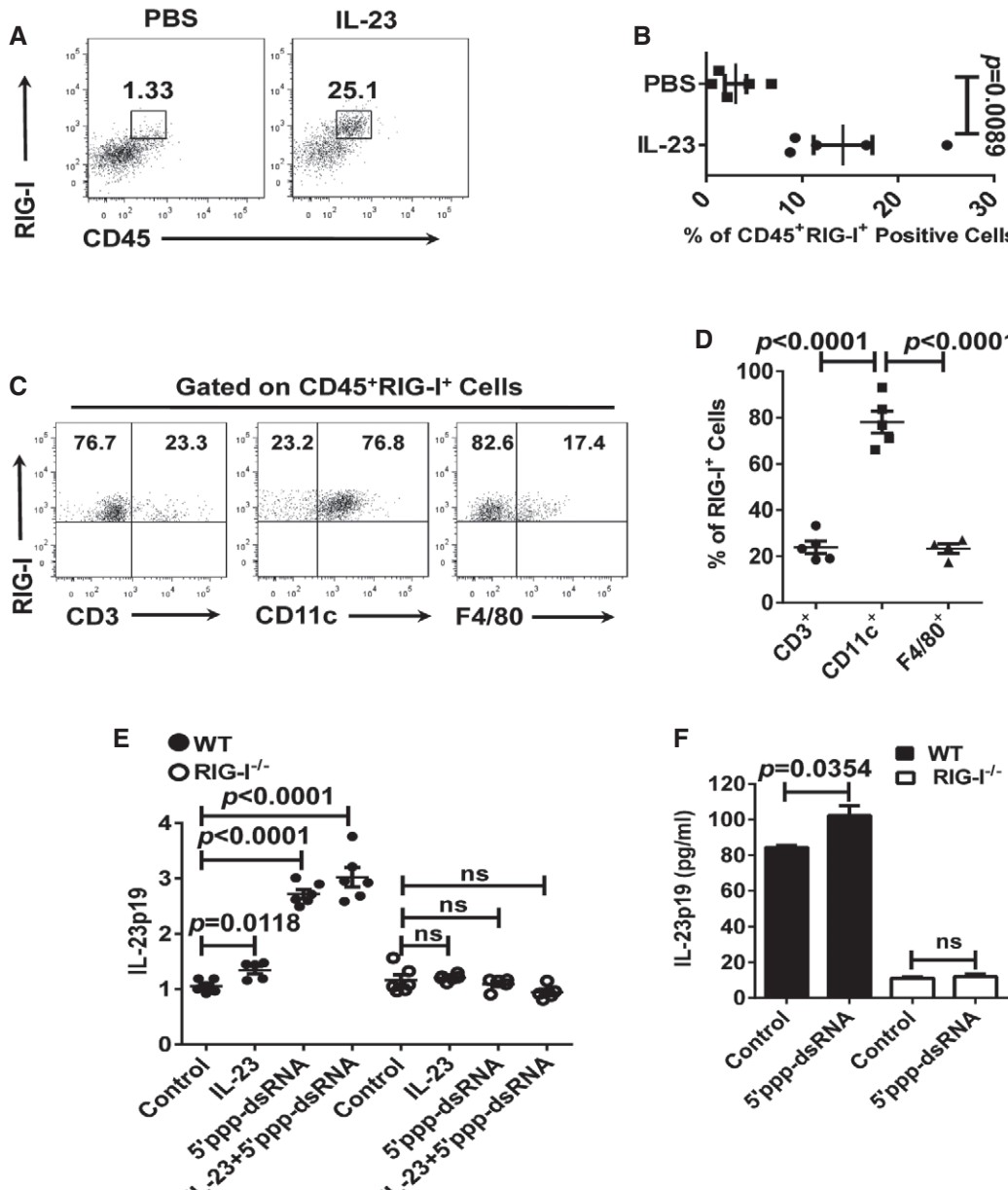

**Figure 6.  RIG-I expression in DCs is critical for IL-23 production.**

A  Representative flow cytometry analysis of the frequency of CD45⁺RIG-I⁺ cells in WT ears treated with PBS or IL-23 (n = 5).

B  Statistical analysis of the results in (A). Significant differences are indicated: two-tailed Student's *t*-test, n = 5 per group (mean ± SEM).

C  Representative flow cytometry analysis of the frequency of RIG-I⁺CD3⁺, RIG-I⁺CD11c⁺, and RIG-I⁺F4/80⁺ cells in ears treated with IL-23 (n = 4–5).

D  Statistical analysis of the results in (C). Significant differences are indicated: two-tailed Student's *t*-test, n = 4–5 per group (mean ± SEM).

E  qPCR analysis of IL-23p19 mRNA expression of cultured WT BMDCs and RIG-I⁻/⁻ BMDCs treated with indicated stimulation for 24 h. qPCR values expressed as the ratio of mRNA to β-actin, relative to negative control, and indicated as fold change. Significant differences are indicated: two-tailed Student's *t*-test, n = 5–6 per group (mean ± SEM).

F  ELISA detection of IL-23p19 protein levels in supernatants of cultured WT BMDCs and RIG-I⁻/⁻ BMDCs treated with indicated stimulation for 48 h. Significant differences are indicated: two-tailed Student's *t*-test, n = 3 per group (mean ± SEM).

indicating that the antiviral signaling pathway may be involved in its etiologic mechanism. In the present study, we provide the first direct evidence that microbiota play a key role in elicitation of psoriasis-like skin disease induced by IL-23. We found that the intradermal injection of IL-23 almost failed to induce psoriasis-like skin

inflammation in GF conditions, suggesting a possible role of microbiota in pathophysiology of IL-23-induced skin inflammation in mice.

RIG-I is generally known as a major sensor of viruses by recognizing the 5′ppp-dsRNA and is suggested as a psoriasis susceptibility

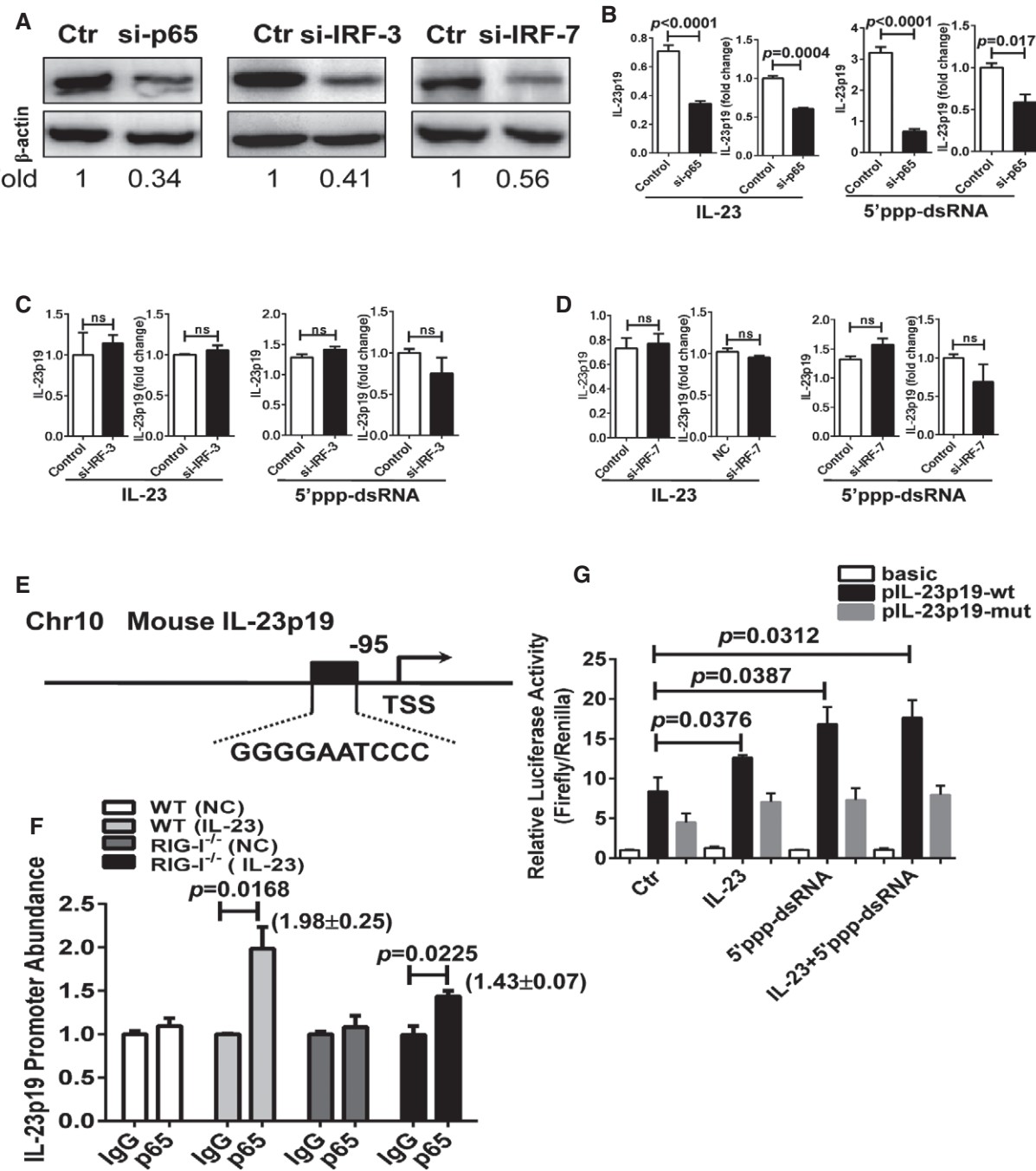

**Figure 7. NF-κB signaling is required for the expression of IL-23p19 in DCs.**

A    BMDCs were transfected with scramble siRNA (Ctr), p65 siRNA (si-p65), IRF-3 siRNA (si-IRF-3), or IRF-7 siRNA (si-IRF-7). Cell lysates were immunoblotted with anti-p65, anti-IRF-3, anti-IRF-7, or anti-β-actin Ab. Values were expressed as fold changes relative to controls and normalized to β-actin.

B–D    qPCR and ELISA analysis of IL-23p19 expression in BMDCs transfected with scramble siRNA (Ctr), p65 siRNA (si-p65) (B), IRF-3 siRNA (si-IRF-3) (C), or IRF-7 siRNA (si-IRF-7) (D), and stimulated with IL-23 or 5′ppp-dsRNA for 30 or 36 h. Significant differences are indicated: two-tailed Student's t-test, n = 3–6 per group (mean ± SEM).

E    The schematic diagram shows one potential binding site of p65 in the putative promoter element of mouse IL-23p19.

F    p65 was immunoprecipitated from BMDCs stimulated with IL-23. Immunoprecipitates were assayed for the enrichment of IL-23 promoter. Significant differences are indicated: two-tailed Student's t-test, n = 3–4 per group (mean ± SEM).

G    Luciferase activity in lysates of RAW264.7 cells transfected with luciferase reporter plasmids of pGL3-basic empty vector (basic), IL-23p19 promoter (pIL-23p19-wt), or IL-23p19 promoter with mutation on the predicted NF-κB binding site (pIL-23p19-mut), unstimulated or stimulated with IL-23 or/and 5′ppp-dsRNA. Results are presented as the ratio of firefly luciferase to *Renilla* luciferase activity, relative to that of unstimulated RAW264.7 cells transfected with pGL3-basic empty vector. Significant differences are indicated: two-tailed Student's t-test, n = 3–4 per group (mean ± SEM).

Source data are available online for this figure.

gene by genome-wide association studies (Tsoi et al, 2012). To date, little is known about the function of RIG-I in the development and progression of psoriasis. We detected high levels of RIG-I expression in lesional skin of psoriasis patients and IL-23-induced psoriasis mouse model. We further demonstrated a close correlation between RIG-I expression and acanthosis in the psoriatic lesions of patients. We speculate that it is possible that high levels of RIG-I expression are triggered by excessive IL-23 in psoriatic lesions as we have found that IL-23 is capable of stimulating RIG-I expression in DCs. A very interesting study reported that Th17 cells strongly produce IL-29, which increases expression levels of antiviral proteins in psoriatic lesions (Wolk et al, 2013). Our study here suggests a potential antiviral function of IL-23 acting through increasing expression of RIG-I sensor of viruses in psoriasis. By using the RIG-I$^{-/-}$ mice, we define that the RIG-I-mediated antiviral signaling is indispensible for experimental psoriasis in IL-23-induced model. Although injecting IL-23 might induce atopic dermatitis-like inflammation rather than psoriasis-like inflammation in CCR2-deficient mice (Bromley et al, 2013), the robust Th1, Th2, Th17 activation, which is similar to human atopic dermatitis, is seen in IL-23-injected mice (Ewald et al, 2017). The IL-23-induced mouse model has been traditionally considered to resemble human psoriasis by exhibiting largest transcriptomic homology with human psoriasis among existing "psoriasis-like" mouse models (Suarez-Farinas et al, 2013), and direct administration of IL-23 results in erythema, mixed dermal infiltrate and epidermal hyperplasia associated with parakeratosis (Chan et al, 2006). What's more, we also confirm that RIG-I is a critical factor for development of psoriasis pathology in IMQ-induced mouse model. For a complementary approach, we confirmed that administration of 5′ppp-dsRNA, which is a synthetic specific ligand for RIG-I, is capable to directly trigger psoriasis-like skin inflammation, highlighting that the activation of RIG-I antiviral signaling is one of initiating factors in psoriasis. Furthermore, we found that RIG-I exerts function in CD11c$^+$ DCs to regulate the pathogenesis of psoriasis. However, we did not detect a major effect of RIG-I on radio-resistant cells resident in skin such as keratinocytes.

DCs are known to be very central pathogenic mediator in psoriasis by activating effector T cells and amplifying immune responses. CD11c$^+$ dermal DCs are the major DC population accumulating in psoriatic lesions, and their numbers are reduced after all successful treatments (Lowes et al, 2014). Very interestingly, ablation of nerve-derived signals results in decreased DCs, IL-23 expression and improved acanthosis (Ostrowski et al, 2011), and skin TRPV1$^+$Nav1.8$^+$ nociceptors, by interacting with dermal DCs, drive IL-23 production in IMQ-induced psoriasis-like skin disease, strongly supporting the notion that these dermal DCs are very essential skin IL-23 producers in the pathophysiology of psoriasis (Riol-Blanco et al, 2014). Although pDCs have been reported to exacerbate IMQ-induced psoriasis-like skin disease by producing IFN-α (Takagi et al, 2016), we observed that RIG-I is mainly expressed by CD11c$^+$ DCs and its activation triggers IL-23 expression and psoriasis-like skin inflammation. It is well established that CD11c$^+$ DCs exert functions by presenting antigen and producing IL-23 and IL-12 to activate and/or differentiate naïve T cells to Th17 and Th1 subsets (Zaba et al, 2007). The IL-23/Th17 axis has been regarded as a key master pathway in psoriasis and also enhances the development of several other models of autoimmune disease

including experimental allergic encephalomyelitis (EAE; Chen et al, 2006), colitis (Sivanesan et al, 2016), and diabetes (Bellemore et al, 2016). Furthermore, clinical improvement in psoriasis with specific targeting of IL-23 has been verified (Kopp et al, 2015). Originally, IL-23 was reported to be in response to pathogens including certain bacteria and viruses or their components (McKenzie et al, 2006). Poly(I:C), which serves as TLR3 and RLR agonist, has been shown to enhance the releasing of IL-23 in keratinocytes (Ramnath et al, 2015). In line with these findings, we have identified that 5′ppp-dsRNA, which is a specific ligand for RIG-I, has the ability to induce IL-23 production by CD11c$^+$ DCs in mice. This pathway in CD11c$^+$ DCs may drive the "quiescent" autoimmune T cells into activated pathogenic effectors.

In addition, how exactly the downstream pathways of RIG-I are linked to IL-23 production is largely unknown. IRF-3/7 and NF-κB are the main downstream signaling pathways of RIG-I (Loo & Gale, 2011). Activation of IRF-3/7 leads to type I interferon production and IFN-α produced mainly by pDCs, contributing to initiate psoriasis (Nestle et al, 2005). Increasing evidence indicates that genetic mutations in genes such as TNFAIP3, TNIP1, NFKBIA, TRAF3IP2, and CARD14 result in an impaired negative regulation of NF-κB proinflammatory activity leading to psoriasis (Nair et al, 2009; Huffmeier et al, 2010; Strange et al, 2010; Jordan et al, 2012). Indeed, the NF-κB inhibition as a therapeutic strategy in psoriasis has been identified (Andres et al, 2013). Moreover, c-Rel, which is a member of NF-κB signaling pathway, plays essential role in TLR-induced IL-23 expression in CD11c$^+$ DCs (Carmody et al, 2007). Consistent with these findings, we clarified that IL-23 production mediated by RIG-I was primarily through NF-κB rather than IRF-3/7 signaling pathway.

Despite that both of the mouse models in our research are acute models of skin inflammation, the phenomena in chronic models need to be further verified, and the precise underlying mechanism requires further studies, the pathological effector function of RIG-I in skin provides an additional novel insight into the pathomechanism of psoriasis. Here, we assume that only IL-23 injection is not sufficient to sustain the endogenous IL-23 production without the RIG-I activation by viruses in GF conditions. RIG-I serves as both a downstream target and a regulator of IL-23. The up-regulation of RIG-I and its innate activation contribute to endogenous IL-23 production, subsequently driving immune responses leading to initiation of psoriasis in susceptible individuals, indicating that RIG-I is probably a critical trigger rather than a regulator in remission. Thus, our research suggests that RIG-I activation in CD11c$^+$ DCs is an important upstream initiator.

Together with previous reports (Wohn et al, 2013; Riol-Blanco et al, 2014), we propose a pathogenic model of psoriasis. In genetically predisposed individuals, the virus infection triggers the activation of TLR-7/8 and/or RIG-I antiviral signaling and subsequently induces IL-23 expression in the CD11c$^+$ DCs via the NF-κB pathway, and genetic mutations in genes such as TNFAIP3, TNIP1, NFKBIA, TRAF3IP2, and CARD14 result in an impaired negative regulation of NF-κB proinflammatory activity (Nair et al, 2009; Genetic Analysis of Psoriasis et al, 2010; Huffmeier et al, 2010; Jordan et al, 2012), thereby leading to psoriasis (Fig EV5). In summary, the present study links RIG-I, an important psoriasis susceptibility gene, with the abnormal activation of NF-κB and excessive production of IL-23 in psoriatic lesions and suggests that

RIG-I, a major cytoplasmic sensor of antiviral immunity, is one of the dominant pathological factors in psoriasis. Our findings also suggest a novel therapeutic strategy by targeting RIG-I activation in CD11c$^+$ DCs in patients with psoriasis.

# Materials and Methods

### Mice

C57BL/6J mice (stock number: 000664) were purchased from The Jackson Laboratory (Bar Harbor, ME). RIG-I$^{-/-}$ mice were kindly provided by Dr. Zhugang Wang (Shanghai Jiao Tong University School of Medicine, Shanghai, China). IL-23$^{-/-}$ mice were kindly provided by Prof. Chen Dong (Institute for Immunology, Tsinghua University, Beijing, China). Mice were bred and maintained under specific pathogen-free (SPF) conditions. Germ-free (GF) C57BL/6J mice were placed and raised in special sterile, flexible plastic gnotobiotic isolators and fed with irradiated standard chow and water (Shanghai SLAC Laboratory Animal Center, Chinese Academy of Sciences). Age- and sex-matched mice at 8–12 weeks of age were randomly used for all experiments in compliance with the National Institutes of Health Guide for the Care and Use of Laboratory Animals with the approval (SYXK-2003-0026) of the Scientific Investigation Board of Shanghai Jiao Tong University School of Medicine, Shanghai, China. At least six mice in total for each experimental were used to allow reliable estimation of within-group variability. To ameliorate any suffering of mice observed throughout these experimental studies, mice were euthanized by $CO_2$ inhalation.

### IL-23-induced mouse model of psoriasis

Mouse ears were injected intradermally with 500 ng recombinant mouse IL-23 (IL-23; R&D Systems, #1887-ML-010) dissolved in 25 μl PBS into one ear and 25 μl PBS into the contralateral ear. Injections were continued every other day for eight times. Ears were collected for analysis on day 16.

### In vivo administration of RIG-I-specific ligand

The 5′ppp-dsRNA/LyoVec (InvivoGen, #tlrl-3prnaclv-100) is a specific ligand for RIG-I. Mouse ears were injected intradermally with 3 μg 5′ppp-dsRNA/LyoVec into one ear and 3 μg 5′ppp-dsRNA Control/Lyovec (InvivoGen, #tlrl-3prnalv-100) into the contralateral ear. Injections were continued every other day for 8 times. Ears were collected for analysis on day 16.

### Human subjects

Psoriatic skin samples were obtained by punch biopsy under local lidocaine anesthesia. Normal adult human skin specimens were taken from healthy undergoing plastic surgery. The fresh tissue samples were frozen in liquid nitrogen and stored at −80°C. Patient information is included in Table EV1. All individuals provided informed consent. The study was performed in accordance with the Declaration of Helsinki Principles and approved by the Research Ethics Board of Shanghai Tenth People's Hospital, Tongji University

School of Medicine, Shanghai, Xiangya Hospital, Central South University, Hunan, China.

### RNA extraction, reverse transcription, and qPCR

Total RNA was extracted from skin biopsies or cultured dendritic cells (DCs) using the TRIzol reagent (Invitrogen, #15596-026), and NanoDrop spectrophotometer (ND-1000) was used for RNA quality control. cDNA was synthesized using SuperScript First-Strand Synthesis System (Invitrogen, #1209992). qPCR was carried out with the FastStart Universal SYBR Green Master (Roche, #04913914001) in a ViiA 7 Real-Time PCR System (Applied Biosystems). The relative expression of target genes was confirmed using quantity of target gene/quantity of β-actin. Primer sequences are listed in Table EV2.

### Western blotting

Mouse ears or cultured DCs were lysed in radio immunoprecipitation assay buffer supplemented with protease and phosphatase inhibitor cocktail (Thermo Scientific, #78440). Mouse anti-β-actin antibody (Ab) (1:2,000; Cell Signaling Technology, #3700S), rabbit anti-RIG-I Ab (1:1,000; Abcam, #ab45428), rabbit anti-p-IκBα Ab (1:1,000; Cell Signaling Technology, #2859), HRP-labeled goat anti-mouse IgG(H+L) (1:2,000; Beyotime, #A0216), and HRP-labeled goat anti-rabbit IgG(H+L) (1:1,000; Beyotime, #A0208) were used. The signal was detected with ECL Western Blotting Substrate (Thermo Scientific, #34095) and GE ImageQuant LAS 4000 (GE Healthcare) or Amersham Imager 600 (GE Healthcare). Images have been cropped for presentation.

### Histological analysis and immunohistochemistry

After treatment with IL-23, the mouse ear skin was fixed in formalin and embedded in paraffin. Sections (6 μm) were stained with hematoxylin and eosin (H&E). Epidermal hyperplasia (acanthosis) was assessed by using average length of three times of measures between the basement membrane and the stratum corneum. For immunohistochemistry, RIG-I expression, Ki67 expression, or CD3 expression was evaluated in skin sections using rabbit anti-RIG-I Ab (1:100 dilution, Abcam, #ab45428), anti-mouse Ki67 Ab (1:100 dilution, Cell Signaling Technology, #12202), or anti-CD3 Ab (1:100 dilution, Abcam, #ab16669), respectively, following the manufacturer's instructions. To quantify the expression level of RIG-I, histochemistry score (H score) was applied and double-blind method was taken to evaluate the positive cases, where both the intensity and the percentage of positivity were considered using the following formula: H score = 3 × (strong intensity) × % + 2 × (moderate intensity) × % + 1 × (mild intensity) × %. For counting dermal infiltrating cells or Ki67$^+$ cells, three areas in three sections of each sample were randomly taken, in which the number of infiltrating cells or Ki67$^+$ cells was calculated.

### Immunofluorescence staining

Frozen cryosections of mice ear skin with the indicated genotype or treatment were fixed in ice-cold acetone for 15 min before staining and then incubated in blocking buffer (1% BSA/PBS) for 1 h. The

slices were stained with anti-CD11c Ab (Abcam, #33483) with a 1:20 dilution and incubated for 48 h at 4°C. Thereafter, the slices were rinsed three times in PBS and incubated in fluorochrome-conjugated secondary Ab (Abcam, #173003) with a 1:500 dilution for 1 h at RT in the dark. DAPI (BD, #564907) was used for nuclei staining with a 1:2,000 dilution at RT for 5 min. After being washed in PBS and air drying, the slices were mounted with fluorescent mounting medium (Dako, #S3023) and examined by confocal microscope (Leica, TCS SP8).

### ELISA

The supernatants of ear skin homogenate or cell culture supernatants were collected for cytokine evaluation. Quantitative analysis of IL-23p19 and IL-17 was performed by enzyme-linked immunosorbent assay according to the manufacturer's instructions (R&D, #M2300, #M1700).

### Generation of bone marrow chimeric mice

Bone marrow cells were prepared from WT or RIG-I$^{-/-}$ donor mice and adoptively transferred into lethally irradiated (950 rad) RIG-I$^{-/-}$ or WT mice (8-week-old, $1 \times 10^7$ per mouse) in two divided doses. After 8 weeks, the chimeric mice were subjected for psoriasis mouse model induction.

### Isolation of cells from ears

Ears were removed and cut into pieces. The cells were separated following incubation with collagenase type IV (2 mg/ml; Gibco, #17104-019) and DNase (Sigma, #10104159001) for 2 h at 37°C by gently shaking. Subsequently pass through 70-μm nylon mesh (BD Biosciences, #352350) to remove undigested skin. After centrifugation at 300 $g$ for 8 min, the supernatant was aspirated and the pellet was resuspended.

### Flow cytometric analysis

Single-cell suspensions from IL-23-induced psoriasiform ears or normal ears were prepared in PBS. For analysis of surface markers, cells were stained in PBS containing 0.5% (wt/vol) BSA with anti-mouse CD45 Ab (BD Biosciences, #560510), anti-mouse CD11c Ab (eBioscience, #25-0114-82), anti-mouse F4/80 Ab (eBioscience, #12-4801-82), and anti-mouse CD3ε Ab (eBioscience, #17-0031-83) for 30 min at 4°C protected from light. For intracellular staining, cells were fixed and permeabilized with BD Cytofix/Cytoperm (BD Biosciences, #51-2090KZ) and were stained with anti-RIG-I Ab (Abcam, #ab45428) and incubated for 30 min at 4°C. Thereafter, the cells were rinsed three times in PBS and incubated in goat anti-rabbit fluorochrome-conjugated secondary antibody (Invitrogen, #A11034) for 30 min at 4°C in the dark. Finally, cells were washed, resuspended, assayed with BD FACSCanto II Flow Cytometer, and analyzed with FlowJo software.

### Bone marrow dendritic cell culture

Bone marrow was flushed from tibias and femurs of WT mice or RIG-I$^{-/-}$ mice and resuspended at $1 \times 10^6$/ml in RPMI medium

1640 basic (Gibco, #C22400500BT) supplemented with β-mercaptoethanol (Gibco, #21985), 10% FBS (Gibco, #10437-028), 10 ng/ml GM-CSF (Peprotech, #AF-315-03), and 10 ng/ml IL-4 (Peprotech, #AF-214-14). Cells were incubated at 37°C with 5% $CO_2$ and cultured for 6 days with media changes on day 3 and day 5.

### Chromatin immunoprecipitation assay

Chromatin immunoprecipitation (ChIP) assays were performed using the SimpleCHIP enzymatic chromatin immunoprecipitation kit (Cell Signaling Technology, #9002) according to the manufacturer's protocol with minor modifications. In brief, the cells were harvested and cross-linked with 1% (v/v) formaldehyde for 10 min at RT. Subsequently, nuclei were isolated by the lysis of cytoplasmic fraction and chromatin was digested into fragments of 150–900 bp by micrococcal nuclease (400 gel units) for 20 min at 37°C, followed by ultrasonic disruption of the nuclear membrane using a standard microtip and a Branson W250D Sonifier (four pulses, 60% amplitude, duty cycle 40%). The sonicated nuclear fractions were divided for input control and for overnight incubation at 4°C with 5 μg either anti-p65 Ab (Cell Signaling Technology, #8242) or the negative control IgG. After incubation with 30 μl of ChIP grade protein G-agarose beads for 2 h at 4°C, the antibody–protein–DNA complexes were then eluted from the beads and digested by proteinase K (40 μg) for 2 h at 65°C, followed by spin column-based purification of the DNA. Finally, genomic DNA recovered from the ChIP assays was qPCR amplified with primers specific to the p65-binding elements of the IL-23p19 promoter region. The specificity of the primer set was verified by analyzing the dissociation curve of each gene-specific PCR product. Primer sequences are listed in Table EV2.

### RNA interference

Custom and chemically modified small interfering RNA (siRNA) was designed to target mouse p65 (GenePharma), IRF-3, and IRF-7. Non-specific siRNA duplex served as control (GenePharma). To knock down p65, IRF-3, or IRF-7 *in vitro*, cultured BMDCs were transfected with siRNA-1822 (p65, 45 pmol/ml), siRNA-744 (IRF-3, 45 pmol/ml), or with siRNA-1168 (IRF-7, 45 pmol/ml) by Lipofectamine 3000 Transfection Reagent (Invitrogen, #L3000-15). siRNA sequences are listed in Table EV2.

### Luciferase reporter assay

PGL3-basic vector (Promega, #E1751) was used to clone the promoter of IL-23p19. Site-specific mutant was generated by PCR. RAW264.7 cells, which were provided from the cell bank of Chinese Academy of Sciences, were seeded in a 48-well plate with a density of $1.5 \times 10^5$/well 1 day before transfection, and then, each well was transfected with a mixture of 250 ng pGL3 luciferase vector and 25 ng pRL-TK renilla vector using Lipofectamine 3000 Transfection Reagent (Invitrogen, #L3000-15). Twelve hours post-transfection, cells were treated with IL-23 (100 ng/ml) or/and 5′ppp-dsRNA (1 μg/ml) for another 24 h before lysis, and luciferase activity was measured on a microplate reader (Berthold, TriStar LB941) by using the Dual-Luciferase Reporter Assay System (Promega, #E1910). The ratio of firefly luciferase to *Renilla*

## The paper explained

### Problem

Psoriasis is a chronic inflammatory disease of the skin affecting 1–3% of the general population. Its etiology remains unclear.

### Results

A dysregulation in the antiviral immune responses of hosts through the innate pattern recognition receptors might initiate chronic skin inflammation in psoriasis. Specifically, RIG-I is found to be a key mediator in psoriasis based on its role in sensing virus infection and promoting IL-23 production.

### Impact

This work revises the current understanding of psoriasis etiology and suggests a novel therapeutic strategy based on targeting RIG-I activation in patients with psoriasis.

luciferase was calculated for each well. Primer sequences are listed in Table EV2.

### Statistical analysis

The data were analyzed with GraphPad Prism 5 and are presented as the mean ± SEM. Student's *t*-test was used when two conditions were compared, and analysis of variance (ANOVA) with Bonferroni or Newman–Keuls correction was used for multiple comparisons. Simple linear regression model was used to analyze the correlation between RIG-I expression and H score. Probability values were indicated and of < 0.05 were considered significant; two-sided Student's *t*-tests or ANOVA was performed. ns, not significant. Error bars depict SEM.

**Expanded View** for this article is available online.

## Acknowledgements

This work is supported by grants from the National Natural Science Foundation of China (Key Program: 31330026, General Program: 31570922 and Key Program: 81430033), the Leading Academic Discipline Project of the Shanghai Municipal Education Commission (16XD1402500), the Key Program of the Shanghai Municipal Education Commission (15431900800), the National Program on Key Basic Research Project (973 Program, 2014CB541905), the National Natural Science Foundation of China (Youth Program: 81502723 and 31500730).

## Author contributions

HuiyuanZ and HonglinW designed the research and analyzed the data. HonglinW supervised the research and wrote the manuscript. HuiyuanZ conducted most of the experiments. YG helped with mouse breeding and mouse model construction. FL offered help in luciferase reporter assays. YS helped with cell isolation and flow cytometry. QY offered helped in RNA interference. JB, ZX, ZL, WC, FK, LZ, HongZ, and HongW helped with the experimental details. ZW, HongxinZ, GW, and XC provided transgenic mouse and human skin samples. FG, BS, and CL helped with data discussion. We are very grateful for the kind offer of IL-23$^{-/-}$ mice from Prof. Chen Dong (Institute for Immunology, Tsinghua University School of Medicine, Beijing, 100062, China).

## Conflict of interest

The authors declare that they have no conflict of interest.

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
