## [Review Process File · EMBO Molecular Medicine]

RIG-I Antiviral Signaling Drives Interleukin-23 Production and Psoriasis-like Skin Disease

Huiyuan Zhu, Fangzhou Lou, Qianqian Yin, Yuanyuan Gao, Yang Sun, Jing Bai, Zhenyao Xu, Zhaoyuan Liu, Wei Cai, Fang Ke, Lingyun Zhang, Hong Zhou, Hong Wang, Gang Wang, Xiang Chen, Hongxin Zhang, Zhugang Wang, Florent Ginhoux, Chuanjian Lu, Bing Su, and Honglin Wang

Corresponding author: Honglin Wang, Shanghai Jiao Tong University School of Medicine

Review timeline:

Submission date:	05 September 2016
Editorial Decision:	28 November 2016
Revision received:	11 January 2017
Editorial Decision:	24 February 2017
Revision received:	28 February 2017
Editorial Decision:	01 March 2017
Revision received:	02 March 2017
Accepted:	03 March 2017

Transaction Report:

Editor: Roberto Buccione

1st Editorial Decision

28 November 2016

Thank you for the submission of your manuscript to EMBO Molecular Medicine. We have now heard back from the Reviewers whom we asked to evaluate your manuscript.

I again sincerely apologise for the very significant delay in reaching a decision on your manuscript. In this case, we first experienced significant difficulties in securing expert and willing Reviewers. I eventually only managed to secure two reviewers. Further to this, the evaluations were delivered with much regrettable delay.

I am proceeding based on the two evaluations obtained so far as further delay cannot be justified. However, I have also consulted with an external expert to inform my final decision.

As you will see, reviewer 1 is quite negative and while not very detailed, notes that the main conclusions are not supported by the data. Reviewer #2 is more positive, but also raises a similar basic concern and asks for further experimentation that might be unfeasible within a reasonable time frame (about 3 months) and, most importantly, with no guarantee of a successful outcome. After further discussion with my colleagues and the advisor, we agreed a revision would be need to be quite extensive and potentially requiring a significant length of time, with an inevitably uncertain outcome.

Given these fundamental concerns and the overall lack of enthusiasm by the Reviewers and our

advisor, I have no choice but to return the manuscript to you at this stage.

I wish to add that, considered the potential interest of these findings, we would have no objection to consider a new manuscript on the same topic if at some time in the near future you have obtained data that would considerably strengthen the message of the study and address the Reviewers' concerns. Please consider, however, that if you were to send a new manuscript this would be treated as a new submission.

At this stage of analysis, though, I am sorry to have to disappoint you. I nevertheless hope, that the reviewer comments will be helpful in your continued work in this area.

***** Reviewer's comments *****

Referee #1 (Remarks):

In this manuscript Zhu et al. claim that the psoriasiform phenotype induced by repeated IL-23 injection is dependent on viral dsRNA triggering of RIG-I with induction of endogenous IL-23. The manuscript has several problems. In particular, the link between IL-23 injection, RIG-I activation and the role of microbiota in the induction of psoriasiform inflammation via endogenous IL-23 is not convincingly demonstrated.

Major issues:

- 1) The fact that psoriasis induced by repetitive IL-23 injection requires endogenous IL-23 is not demonstrated. Does IL-23 injection into IL-23^{-/-} mice induce less acanthosis?
- 2) What is the mechanism that links IL-23 injection to RIG-I activation by viral 5'ppp-dsRNA? IL-23 upregulates RIG-I in SPF mice but not GF mice (Fig. 1G). How is this explained?
- 3) Is there a constitutive lack of RIG-I expression in GF mice or is it a lack of upregulation due to lack of activation in the absence of microbial dsRNA? What is the role of gut versus skin microbiota in this, in other words, can the reconstitution of skin microbiota restore RIG-I expression and development of psoriasiform skin lesions?
- 4) The conclusion that the microbiota is essential in the pathophysiology of the disease cannot be drawn by the data shown.

Minor issues:

- 5) The development of skin acanthosis upon il-23 injection is completely abrogated in GF mice but not in RIG-I^{-/-} mice. In fact, RIG-I^{-/-} mice seem to develop significant acanthosis. What is the reason for this discrepancy?
- 6) In supplementary Figure 2 the authors propose a model in which CD11c DC are activated by viral nucleic acids through both RIG-I and, surprisingly, TLR7/8 which has not been addressed in this study. Would this explain issue (5)?

Referee #2 (Comments on Novelty/Model System):

I think the findings are interesting - but the outcomes are limited in what they present - and they need to validate their findings in a second psoriasis-like actor model - such as Imiquimod, which is also IL-23 dependent. I have noted this in my comments to the authors.

Referee #2 (Remarks):

This is a very interesting paper that presents the argument that activation of RIG-I (which has recently been identified as a psoriasis susceptibility gene) by 5'ppp-dsRNA, directly causes the production of IL-23 and triggers psoriasis-like skin disease in mice. The authors present interesting data in germ free animals suggesting that commensals may contribute to the activation of IL-23 production via DCs via NFκB signaling. The text is extremely well written, and the story well presented. The figures, however, don't always align with the strong interpretation of the data, as the improvement in the skin phenotype appears frequently to be quite modest.

The work would be strengthened if the authors could address the following issues:

Do they see the same results using a second mouse model of acute-initiated psoriasiform skin

inflammation that is IL-23 dependent - i.e. can they demonstrate similar outcomes using imiquimod-elicited changes in their germ free conditions and in their KO model?

There has been much discussion in the skin community that intradermal IL-23 actually better models atopic dermatitis rather than psoriasis - the authors need to accommodate this discussion in their interpretation of their results.

What happens with WT bone marrow is transplanted into the KO mice, and then treated with either intradermal-IL-23 or imiquimod?

The outcomes the authors present currently are limited, additional outcomes would enhance their findings and interpretation. What happens to skin IL-17A? What happens to the numbers and activation of DCs and T cells in the skin?

The method for analyzing acanthosis seems odd. The images presented show that there is still significant skin inflammation - i.e. the acanthosis of the skin is still present. Can they measure the intrafollicular epidermal thickness measures using average length (in microns) across the entire skin section (i.e. multiple measures between the basement membrane and the stratum corneum (not including any scale)? They can use Adobe photoshop to do this with the measure tool and calibrating it with a micrometer.

The authors need to discuss Dan Kaplan's recent work showing the nerve-DC-IL-23 interactions are critical for candidiasis - and also put those findings into context with theirs. It's possible that there is a neural influence on their outcomes as well, this has not been discussed.

How do the authors control for changes in ear thickness that simply happen as a result of the mouse ear anatomy and how close the ear skin is to the head? I.e. skin distal to the head is much thinner than ear skin closer to the head and can artificially inflate differences in ear thickness that are independent of the experimental manipulation.

1st Revision - authors' response

11 January 2017

Reviewer #1:

1. The fact that psoriasis induced by repetitive IL-23 injection requires endogenous IL-23 is not demonstrated. Does IL-23 injection into IL-23^{-/-} mice induce less acanthosis?

Re: It is a great question. To follow the reviewer's suggestion, we performed experiments by injecting recombinant IL-23 to ears of IL-23^{-/-} mice. We found that IL-23^{-/-} mice exhibited less severity of psoriasis-like skin inflammation compared with wild-type (WT) mice, including ear thickness and other disease features which is consistent with the study that targeting IL-23 results in clinical improvement in psoriasis (Kopp, Riedl et al., 2015) (Supplementary Fig. 1).

2. What is the mechanism that links IL-23 injection to RIG-I activation by viral 5'-ppp-dsRNA? IL-23 up-regulates RIG-I in SPF mice but not GF mice (Fig. 1G). How is this explained?

Re: We appreciate this constructive question raised by the reviewer. In our study we observed that besides the synthetic ligand of RIG-I 5'-ppp-dsRNA, IL-23 is also able to induce up-regulation of RIG-I, suggesting that IL-23 is possibly involved in host antiviral responses via regulating RIG-I expression (Fig. 3E). We further found that activation of RIG-I was able to induce endogenous IL-23p19 expression via NF-kappa B signaling (Fig. 7B and Fig. 7F). However, the detailed molecular mechanism that links IL-23 to RIG-I activation is subjected to another study.

SPF condition means there is only specific pathogen free but still exists some other microorganisms, suggesting that the presence of microorganism is essential for the activation of RIG-I. By contrast,

there is free of all microorganism in GF conditions, and in such conditions only IL-23 injection is not sufficient to sustain the endogenous IL-23 production without RIG-I activation by viruses. We have discussed it in the section of Discussion. **Page 16, line 6-11.**

3. Is there a constitutive lack of RIG-I expression in GF mice or is it a lack of up-regulation due to lack of activation in the absence of microbial dsRNA? What is the role of gut versus skin microbiota in this, in other words, can the reconstitution of skin microbiota restore RIG-I expression and development of psoriasiform skin lesions?

Re: It is a very nice question. The lack of up-regulation of RIG-I is likely due to lack of RIG-I activation triggered by microbial dsRNA because we observed there was very similar low levels of RIG-I expression in skin of GF and SPF mice when both treated by PBS (Fig. 1G).

It was published that the gut microbiota promotes psoriasis-like skin disease in imiquimod (IMQ)-induced murine model of psoriasis (Zakostelska, Malkova et al., 2016). To address whether skin microbiota (in our study particularly, RNA viruses) is critical for RIG-I expression and development of psoriasis-like skin lesions, we used IMQ to induce skin inflammation in RIG-I^{-/-} mice. We found that the disruption of RIG-I-mediated RNA virus infection attenuated psoriasis-like skin inflammation in RIG-I^{-/-} mice (Supplementary Fig. 3). Moreover, the IMQ treatment was able to up-regulate RIG-I expression in skin of WT mice (Figure 3F). Together, our data suggest that not only 5'-ppp-dsRNA, but also IL-23 and IMQ treatment are able to up-regulate RIG-I expression, and RIG-I-mediated antiviral signaling is critical for the development of psoriasis-like skin disease in IL-23- and IMQ-induced mouse models.

4. The conclusion that the microbiota is essential in the pathophysiology of the disease cannot be drawn by the data shown.

Re: The reviewer is right that we can not draw the conclusion that the microbiota is essential in the pathophysiology of psoriasis. However, in the current study we suggest that the RIG-I-mediated antiviral signaling triggers IL-23p19 expression and promotes psoriasis-like skin disease in both IL-23- and IMQ-induced mouse models of psoriasis (Fig. 4 and Supplementary Fig. 3). We have modified our writings in the revised manuscript accordingly. Page 13, line 12-13.

5. The development of skin acanthosis upon IL-23 injection is completely abrogated in GF mice but not in RIG-I^{-/-} mice. In fact, RIG-I^{-/-} mice seem to develop significant acanthosis. What is the reason for this discrepancy?

Re: We thank this very helpful question raised by the reviewer. We propose the fact that the RIG-I^{-/-} mice had some disease characteristics might be due to the RIG-I pathway is a critical, but not the only pathway in the pathogenesis of psoriasis as stimulation of the antiviral pattern-recognition receptors such as TLR7/TLR8 with IMQ, a synthetic agonist, is sufficient to trigger psoriasis-like skin inflammation in mice (van der Fits, Mourits et al., 2009).

6. In supplementary Figure 2 the authors propose a model in which CD11c DCs are activated by viral nucleic acids through both RIG-I and, surprisingly, TLR7/8 which has not been addressed in this study. Would this explain issue (5)?

Re: We have followed the reviewer's suggestion. We used IMQ to induce skin inflammation in RIG-I^{-/-} mice. We found that the disruption of RIG-I-mediated RNA virus infection slightly, but

significantly attenuated psoriasis-like skin inflammation in RIG-I^{-/-} mice (Supplementary Fig. 3).

Reviewer #2 (Remarks):

This is a very interesting paper that presents the argument that activation of RIG-I (which has recently been identified as a psoriasis susceptibility gene) by 5'ppp-dsRNA, directly causes the production of IL-23 and triggers psoriasis-like skin disease in mice. The authors present interesting data in germ free animals suggesting that commensals may contribute to the activation of IL-23 production via DCs via NFκB signaling. The text is extremely well written, and the story well presented. The figures, however, don't always align with the strong interpretation of the data, as the improvement in the skin phenotype appears frequently to be quite modest.

The work would be strengthened if the authors could address the following issues:

1. Do they see the same results using a second mouse model of acute-initiated psoriasiform skin inflammation that is IL-23 dependent - i.e. can they demonstrate similar outcomes using imiquimod-elicited changes in their germ free conditions and in their KO model?

Re: This is a very helpful suggestion. It was just published that the psoriasis-like skin inflammation induced by IMQ is attenuated in GF conditions (Zakostelska et al., 2016). We have followed the reviewer's comments by treating IMQ in RIG-I^{-/-} mice in SPF conditions. We found that the RIG-I^{-/-} mice showed amelioration of psoriasis-like skin inflammation when treated with IMQ (Supplementary Fig. 3). These mice had thinner ear thickness, less ki67 positive staining and decreased expression levels of IL-23p19 and IL-17 (Supplementary Fig. 3). Additionally, we found that repeated injections of ears with IL-23 induced a profound increase of IL-17 protein levels in the SPF mice, whereas IL-23 injection did not induced IL-17 expression in the GF mice (Fig. 11).

2. There has been much discussion in the skin community that intradermal IL-23 actually better models atopic dermatitis rather than psoriasis - the authors need to accommodate this discussion in their interpretation of their results.

Re: We have followed the reviewer's helpful suggestion and discussed it in the section of Discussion. Page 14, line 3-6.

3. What happens with WT bone marrow is transplanted into the KO mice, and then treated with either intradermal-IL-23 or imiquimod?

Re: In fact, we have showed these data in Supplementary Fig. 4. When RIG-I^{-/-} mice were reconstituted with WT bone marrow, these mice exhibited comparable levels of severity of psoriasis-like skin inflammation compared with WT mice.

4. The outcomes the authors present currently are limited, additional outcomes would enhance their findings and interpretation. What happens to skin IL-17A? What happens to the numbers and activation of DCs and T cells in the skin?

Re: We followed the reviewer's comments, and investigated the IL-17 protein levels in skin by ELISA, and confirmed that after repeated injections of IL-23, the WT mice had a dramatic increase in IL-17A expression when compared with the PBS treated group (Supplementary Fig. 2D). Meanwhile, the RIG-I^{-/-} mice only slightly elevated compared with WT mice (Supplementary Fig.

2D). By performing staining we found there was a pronounced increase in the expression of CD3 and CD11c in WT mice compared to RIG-I^{-/-} mice (Supplementary Fig. 2A-C).

5. The method for analyzing acanthosis seems odd. The images presented show that there is still significant skin inflammation - i.e. the acanthosis of the skin is still present. Can they measure the intrafollicular epidermal thickness measures using average length (in microns) across the entire skin section (i.e. multiple measures between the basement membrane and the stratum corneum (not including any scale)? They can use Adobe photoshop to do this with the measure tool and calibrating it with a micrometer.

Re: We followed the reviewer's comments, and corrected it by measuring the thickness between the basement membrane and the stratum corneum by using average length in microns.

6. The authors need to discuss Dan Kaplan's recent work showing the nerve-DC-IL-23 interactions are critical for candidiasis - and also put those findings into context with theirs. It's possible that there is a neural influence on their outcomes as well, this has not been discussed.

Re: It is a very nice suggestion, and we have discussed it in the discussion. Page 14, line 16-19.

7. How do the authors control for changes in ear thickness that simply happen as a result of the mouse ear anatomy and how close the ear skin is to the head? I.e. skin distal to the head is much thinner than ear skin closer to the head and can artificially inflate differences in ear thickness that are independent of the experimental manipulation.

Re: It is correct that skin distal to the head is much thinner than ear skin closer to the head. To make our data as precise as possible, we choose to measure the same site in the middle of the mouse ears and use the relative value to evaluate ear thickness.

References

Kopp T, Riedl E, Bangert C, Bowman EP, Greisenegger E, Horowitz A, Kittler H, Blumenschein WM, McClanahan TK, Marbury T, Zachariae C, Xu D, Hou XS, Mehta A, Zandvliet AS, Montgomery D, van Aarle F, Khalilieh S (2015) Clinical improvement in psoriasis with specific targeting of interleukin-23. *Nature* 521: 222-6

van der Fits L, Mourits S, Voerman JS, Kant M, Boon L, Laman JD, Cornelissen F, Mus AM, Florenzia E, Prens EP, Lubberts E (2009) Imiquimod-induced psoriasis-like skin inflammation in mice is mediated via the IL-23/IL-17 axis. *J Immunol* 182: 5836-45

Zakostelska Z, Malkova J, Klimesova K, Rossmann P, Hornova M, Novosadova I, Stehlikova Z, Kostovcik M, Hudcovic T, Stepankova R, Juzlova K, Hercogova J, Tlaskalova-Hogenova H, Kverka M (2016) Intestinal Microbiota Promotes Psoriasis-Like Skin Inflammation by Enhancing Th17 Response. *PLoS One* 11: e0159539

2nd Editorial Decision

24 February 2017

Thank you for the submission of your revised manuscript to EMBO Molecular Medicine. I again apologise for the very unusual delay in providing you with a decision.

Unfortunately we could not retrieve an opinion from reviewer 1 notwithstanding repeated emails and phone calls. While we find this inexcusable, we eventually went back to reviewer 2 to ask his/her opinion on your rebuttal to reviewer 1's comments.

Based on the enclosed report from reviewer 2 and his/her statement that your actions on reviewer 1's concerns are satisfactory, I am pleased to inform you that we will be able to accept your manuscript pending the following final amendments:

1) Please take appropriate action on reviewer 2's final requests. I agree that to do so would improve the overall quality of your discussion and presentation

2) Every published paper includes a 'Synopsis' to further enhance discoverability. Synopses are displayed on the journal webpage and are freely accessible to all readers. They include a short description as well as 2-5 one-sentence bullet points that summarise the key NEW findings of the paper. The bullet points should be designed to be complementary to the abstract - i.e. not repeat the same text. We encourage inclusion of key acronyms and quantitative information. Please use the passive voice. Please attach this information in a separate file or send them by email, we will incorporate it accordingly. We also encourage the provision of striking image or visual abstract to illustrate your article. If you do, please provide a jpeg file 550 px-wide x 400-px high.

3) Please upload your figures as individual files

4) We encourage the publication of source data, with the aim of making primary data more accessible and transparent to the reader. Would you be willing to provide a PDF file per figure that contains the original, uncropped and unprocessed scans of all or at least the key gels used in the manuscript and/or source data sets for relevant graphs? The files should be labeled with the appropriate figure/panel number, and in the case of gels, should have molecular weight markers; further annotation may be useful but is not essential. The files will be published online with the article as supplementary "Source Data" files. If you have any questions regarding this just contact me.

5) We now mandate that all corresponding authors list an ORCID digital identifier. You may acquire one through our web platform upon submission and the procedure takes <90 seconds to complete. We also encourage co-authors to supply an ORCID identifier, which will be linked to their name for unambiguous name identification.

Please submit your revised manuscript within two weeks. I look forward to seeing a revised form of your manuscript as soon as possible.

Finally, I hope that the inevitable frustration caused by this delay is somewhat tempered by the positive outcome.

***** Reviewer's comments *****

Referee #2 (Comments on Novelty/Model System):

Thanks for validating your findings in a second model.

Referee #2 (Remarks):

The authors have done what I requested.

Issues still needing a bit of attention are detailed below:

They need to still better discuss the nuances of IL-23 as an AD model vs a psoriasis model and incorporate the recent JACI paper - by Emma Guttman - Ewald et al , 2016

They need to emphasize that their effects hold up across TWO ACUTE models of elicited skin inflammation - IL-23 and imiquimod - and make the wording less strong on Imiquimod being psoriasis - its not - it's a murine model of some aspects of psoriasis-like disease

They need to add a comment that they are unclear what will happen in a chronic mouse model of

psoriasis and its possible the results could differ - reflecting criticality at initiation and perhaps less during reversal and remission

If they are going to quote the Riol Blanco paper - they really should include the first paper that showed this phenomena - the Ostrowski et al JID paper. - which demonstrates disease improvement rather than prevention...and again the authors are encouraged to also include The Dan Kaplan work looking at nerves and DC interactions.

2nd Revision - authors' response

28 February 2017

Reviewer #2 (Comments on Novelty/Model System):

Thanks for validating your findings in a second model.

Referee #2 (Remarks):

The authors have done what I requested.

Issues still needing a bit of attention are detailed below:

1. They need to still better discuss the nuances of IL-23 as an AD model vs a psoriasis model and incorporate the recent JACI paper - by Emma Guttman - Ewald et al, 2016.

Re: We followed the reviewer's comments, and discussed the nuances of IL-23 as an AD model vs a psoriasis model in the section of DISCUSSION (Page 14, line 2-8).

2. They need to emphasize that their effects hold up across TWO ACUTE models of elicited skin inflammation - IL-23 and imiquimod - and make the wording less strong on Imiquimod being psoriasis - its not - it's a murine model of some aspects of psoriasis-like disease.

Re: We are so grateful for the reviewer's suggestion, and we corrected our wording by using "IMQ-induced psoriasis-like mouse model" in the whole manuscript. We also emphasized that the mouse model we used in our research only represented the acute phase of psoriasis (Page 16, line 5-6).

3. They need to add a comment that they are unclear what will happen in a chronic mouse model of psoriasis and its possible the results could differ - reflecting criticality at initiation and perhaps less during reversal and remission.

Re: It is a very nice suggestion, and we have added a comment to clarify that the phenomena in chronic models need to be further investigated (Page 16, line 5-6) and RIG-I is probably a critical trigger rather than a regulator in remission in psoriasis (Page 16, line 14).

4. If they are going to quote the Riol Blanco paper - they really should include the first paper that showed this phenomena - the Ostrowski et al JID paper. - which demonstrates disease improvement rather than prevention...and again the authors are encouraged to also include The Dan Kaplan work looking at nerves and DC interactions.

Re: We have followed the reviewer's helpful suggestions and discussed it in the section of DISCUSSION (Page 14, line 18-19).

3rd Editorial Decision

01 March 2017

Unfortunately, there are a few final items for you to act upon, which we noticed only on your last submission:

1) Thank you for providing source data for some of the figures. I note however, that that there does not appear to be a perfect match between Fig 3 panel E and the corresponding blot in the source data file. Please correct/explain. Also, please provide source data as one file per each figure.

2) As per our Author Guidelines, the description of all reported data that includes statistical testing

must state the name of the statistical test used to generate error bars and P values, the number (n) of independent experiments underlying each data point (not replicate measures of one sample), and the actual P value for each test (not merely 'significant' or 'P < 0.05'). You may wish to collect all P values in a separate table and refer to it in the legends. I am sorry that I omitted to mention this before, but do note that this requirement is clearly mentioned in the author guidelines (<http://embomolmed.embopress.org/authorguide#datapresentationformat>)

3) We suggest that you remove all siRNA and primer sequences from the main text and collect them in a separate table, which you can refer to in the text.

4) Please provide TIFF rather than JPEG files

5) Please try to make the scale bars a bit more clearly visible

6) Please make the supplementary figures Expanded View figures instead (<http://embomolmed.embopress.org/authorguide#expandedview>) and carefully update figure callouts in the manuscript

7) References cannot feature more than 20 authors et al. Please correct.

8) We are still missing the Author Checklist (<http://embomolmed.embopress.org/authorguide>).

Please submit your revised manuscript within two weeks. I look forward to seeing a revised form of your manuscript as soon as possible. The sooner you send as the revised files, the sooner I can accept!

3rd Revision - authors' response

02 March 2017

Thank you so much for your efforts to accept our revised manuscript (**EMM-2016-07027**) for publication in *EMBO Molecular Medicine*. We think that we have satisfactorily amended the manuscript, and loaded all required files to the Journal.

Corresponding Author Name: Honglin Wang

Manuscript Number: EMM-2016-07027